Species diversity of drifting fish eggs in the Yangtze River using molecular identification

Liu Mingdian
Wang Dengqiang
Gao Lei
Tian Huiwu
Liu Shaoping
Chen Daqing
Duan Xinbin duan@yfi.ac.cn lmd800226@163.com
Yangtze River Fisheries Research Institute of Chinese Academy of Fishery Science , Wuhan , Hubei , China
Rahman Mohammad Shamsur
Electronic publication date: 2018 Oct 24
Publication date: 2018
Volume: 6
Electronic Location ID: e5807
Received 2018 Apr 23; Accepted 2018 Sep 21
Copyright: ©2018 Liu et al.
Copyright year: 2018
Copyright holder: Liu et al.
License: This is an open access article distributed under the terms of the Creative Commons Attribution License, which permits unrestricted use, distribution, reproduction and adaptation in any medium and for any purpose provided that it is properly attributed. For attribution, the original author(s), title, publication source (PeerJ) and either DOI or URL of the article must be cited.
License URL: https://creativecommons.org/licenses/by/4.0/

Keywords: Fish egg, Species diversity, Drifting egg, Molecular Identification, Yangtze River

Funding: National Natural Science Foundation of China 51579247 51409280 This work was supported by the National Natural Science Foundation of China (No. 51579247 and No. 51409280). The funders had no role in study design, data collection and analysis, decision to publish, or preparation of the manuscript.

==============================
The dam constructions greatly changed the hydrologic conditions in the Yangtze River, and then significantly affected the spawning activities of indigenous river fish. Monitoring the species composition of drifting eggs during spawning season is important for protection issues. In this study, we have sampled drifting fish eggs in nine locations from 2014 to 2016. Eggs were identified using the mitochondrial cyt b gene sequence. A total of 7,933 fish eggs were sequenced successfully and blasted into the NCBI database. Thirty-nine fish species were identified, and were assigned to four families and two orders. Approximately 64% of the species identified, and 67% of the eggs, were classified in the Family Cyprinidae. Abundance and Shannon–Wiener diversity index of species were higher in the main river than in tributaries of the river. However, tributaries may be important spawning grounds for some fish species. The Jaccard’s similarity index and river-way distances among sampled stations were negatively correlated suggesting the environment shapes species composition in the sampled spawning grounds. These results showed that mitochondrial DNA sequence is a powerful and effective tool for fish egg identification in Yangtze River and these data are useful for conservation efforts.

Introduction

Monitoring fish spawning grounds in natural rivers provides useful information concerning fish habitat, spawning activity, and the effect of anthropogenic stressors. Data regarding spawning species, scale, and time are important for improving management and policy decisions.

The Yangtze River is the third largest river in the word. The section of the river above Yichang is generally called the “upper reaches” and the section from Yichang to Hukou is called the “middle reaches”. In recent years, more dams have been constructed in the river, which seriously affect the spawning environment for fish. Such drastic ecological and environmental changes particularly affect fish that lay drifting eggs. There are approximately four hundred species of fish in the Yangtze River, and of these, approximately thirty species lay drifting eggs. Some of these fish are of commercial importance in China, such as grass carp (Ctenopharyngodon idella Valenciennes, 1844), black carp (Mylopharyngodon piceus Richardson, 1846), silver carp (Hypophthalmichthys molitrix Valenciennes, 1844), and bighead carp (Hypophthalmichthys nobilis Richardson, 1845); these are referred to as the four domestic Chinese carps (FDC) in China. The FDC spawning activities are controlled by temperature and hydrographs (Stanley, 1978; Li et al., 2013), which shape more than 40 spawning grounds within mainstream regions of the Yangtze River. The mean annual egg production was over 100 billion in the 1960s (Yi & Liang, 1964). The construction of the Gezhouba dam in the 1980s, and then the Three Gorges dam in the early 2000s have greatly changed the hydrologic conditions in the middle Yangtze River (Yi, Wang & Yang, 2010). These dam constructions significantly affected the spawning activities of indigenous river fish. The abundance of FDC larvae in the middle Yangtze River had decreased to 105 million in 2005 (Duan et al., 2009), and the spawning time was also delayed by approximately one month (Zhang, Wu & Li, 2012). In addition, there are some species of fish that prefer certain river locations for spawning. For example, the largemouth bronze gudgeon, Coreius guichenoti Sauvage & Dabry de Thiersant, 1874, primarily spawns in the lower of the Jinsha River, which is the main tributary of the Yangtze River (Cao et al., 2007). Impoundment of the Xiangjiaba dam in 2012 has destroyed the spawning ground of this species, and as a result no eggs or larvae have been found since then (Gao, Tang & Chen, 2015). If this species cannot find another suitable spawning ground, it is in danger of extinction (Cheng et al., 2015). Many tributaries of the Yangtze River are also distributed spawning grounds of fish that lay drifting eggs, and some of these fishes are endemic to one or several river sections (Xie et al., 2014; Wan et al., 2011; Wu, Wang & Liu, 2010).

Fishery resources in the Yangtze River have been undergoing a serious recession over the past 30 years (Chen et al., 2009; Huang & Li, 2016). In an attempt to remedy the situation, the government has implemented measures to restore fish resources and the ecosystem. Several parameters include setting up protected regions, closed fishing seasons, releasing artificial fish into the river, and restricting fishing instruments (Chen et al., 2009). Monitoring spawning grounds could assess whether these measures have been successful. During the spawning season there may be more than a dozen or more species of fish spawning in the same location (Xie et al., 2014; Wan et al., 2011; Wu, Wang & Liu, 2010). Different species are associated with different nursery habitats and dispersal during early life history stages, and also have different protection requirements. Thus, quantifying and classifying fish eggs is crucial for conservation, management, and assessment of environmental effects. It is challenging to determine the species of the drifting eggs because most of the fish with this reproductive pattern have similar morphological characters in the early egg stages. In the past, several researchers tried to identify eggs species by size, color, oil droplets (Gao et al., 2010; Cao et al., 2007; Yi, Yu & Liang, 1988), and couldn’t obtain enough diagnosable information at species-level, because the egg size was variable and overlapping among most fishes, and the color was not so clear. Others tried to hatch eggs to identify distinguishable characteristics (Xu et al., 2015; Liu, Li & Liu, 2014); however, it proved difficult to hatch eggs in the field.

Over the past decade, molecular identification methods have been increasingly used to deal within the limitations of morphological identifications for fish eggs. Mitochondrial DNA (mtDNA) DNA gene cytochrome c oxidase subunit I (COI) barcoding was proposed as a global DNA barcoding identification system for animals (Hebert et al., 2003; Hebert, Ratnasingham & Waard, 2003) based on the assumption of a so-called barcoding gap between species that intraspecific COI variation is lower than interspecific COI variation. At present, this approach has been used to the delimitation of various fishes (Hubert et al., 2010; Costa et al., 2012; Landi et al., 2014; Frantine-Silva et al., 2015; Shen et al., 2016, eggs and larvae (Ahern et al., 2018; Burghart et al., 2014; Harada, Lindgren & Hermsmeier, 2015; Lewis et al., 2016; Ardura et al., 2016; Thompson, 2016). A fragment of mtDNA PCR restriction fragment length polymorphism (RFLP) was used for egg identification of European horse mackerel (Nikoletta et al., 2010), the Atlantic cod (Gadus morh ua Linnaeus, 1758) (Bui et al., 2011), Antarctic fish (Fitzcharles, 2012) and pelagic fish in the North Sea (Lelièvre et al., 2012). The mtDNA control region (D-loop) sequence was used to identify marine fish eggs in Taiwan (Shao, Chen & Wu, 2002) and pufferfish in the Sea of Japan for identification of spawning sites (Katamachi, Ikeda & Uno, 2015). The sequence of 16S rRNA gene successfully discriminated the pelagic fish eggs in the western North Pacific (Kawakami, Aoyama & Tsukamoto, 2010). Another mtDNA coding gene, cytochrome b (cyt b), was broadly used as a marker for phylogeny in various taxonomic fishes (e.g.,  Peng, He & Zhang, 2004; Zhang et al., 2010), and also for species identification (Kartavtsev et al., 2007; Zhang et al., 2010). The gene has proven to be highly efficient and reliable for pelagic eggs in southeastern Australia (Neira et al., 2015).

In this study, we have applied cyt b gene for identification of drifting fish eggs sampled from several spawning grounds in the Middle and Upper Yangtze River. The objective was to (i) test identification efficiency of cyt b gene for drifting eggs in the Yangtze River, and (ii) investigate species composition and distribution in different spawning grounds.

Materials and Methods

Sampling location and egg collection

May to July of each year is the spawning season for most fish in the Yangtze River. During this period in 2016, nine stations were set up to collect drifting fish eggs. These were the Yangtze River Panzhihua section (YPZ), the Yangtze River Jiangjin section (YJJ), the Yangtze River Yidou section (YYD), the Yangtze River Jingzhou section (YJZ), the Yangtze River Sanzhou section (YSZ), the Yangtze River Yanwo section (YYW), the Minjiang River Yibin section (MYB), the Chishui River Chishui section (CCS) and the Xiangjiang River Yingtian section (XYT) (Fig. 1). Among these stations, the four stations of YPZ, YJJ, CCS and MYB were located in the upper reaches and the five stations of YYD, YJZ, YSZ, YYW and XYT in the middle reaches of the Yangtze River. Three stations, MYB, CCS and XYT were located in tributary rivers, and the other six stations were located in the main stream of the Yangtze River. Eggs were collected from YJJ in 2014 and 2015, and from YSZ and YYW in 2015. These collections were also used in this study.

Figure 1 Location of nine sample stations and species composition in Family.

1-Gezhouba Dam, 2- Three Gorges Dam, 3- Xiangjiaba Dam.

Collecting nets and methods were described as Duan et al. (2009). Trap nets were put down in water at 6:00 to 9:30 and 16:00 to 19:00 every day during spawning season, and retrieved every 15 min. Each egg was recorded for morphological traits, such as developmental stage and size, and then preserved in a 2 mL EP tube with absolute ethanol.

The sampling work has been approved by the local related management departments named fishing administration supervision and administration station. This study is fully complied with the relevant laws and ethics of the country.

DNA extraction, PCR

Ethanol was decanted off specimens and eggs were soaked in 1.5 mL ddH2O for 1 h. Water was then removed with a pipette, and egg genomic DNA was extracted using an easy-DNA Kit (Omega, Stamford, CT, USA).

Mitochondrial (mt) cyt b gene was used as a molecular marker for species identification. The forward primer was L14322:5′-GAC TTG AAG AAC CAC CGT TGT TAT TCA AC-3′ and the reverse primer H15576: 5′-GCG CTA GGG AGG AAT TTA ACC TCC-3′. PCR reactions had a final volume of 25 µl and contained 2.5 µl 10 ×  PCR Buffer (TaKaRa, Dalian, China), 0.2 µl of 10 mmol/L dNTPs, 1 µL of each 10 µmol/L primers, 0.5 u Taq enzyme (TaKaRa, Dalian, China), 1 µl of template DNA. The PCR reaction cycles were as follows: pre-denaturation at 94 °C for 4 min, 35 cycles of denaturation at 94 °C for 30 s, annealing at 54 °C for 30 s, extension at 72 °C for 90 s, and a final extension at 72 °C for 8 min.

PCR products were checked via 1.5% agarose gel electrophoresis and visualized with ethidium bromide to verify successful amplification. PCR DNA bands with expected size were purified using Cycle-Pure Kit (Omega, Stamford, CT, USA) and sequenced with the same primers as PCR (Tianyi biotech. Co. Ltd, Wuhan, China).

Data analysis

All sequences were aligned and trimmed to the same length using Muscle (Edgar, 2004) in MEGA X (Kumar, Stecher & Li, 2018). The sequence properties of indels, frameshift mutations, in-frame stop codons were careful examined to eliminate presence of pseudogene or nonfunctional sequences. The genetic distance based on Kimura 2-parameter (Kimura, 1980) between the individuals were calculated using MEGA X (Kumar, Stecher & Li, 2018) and a neighbor-joining tree (NJ tree) (Saitou & Nei, 1987) was constructed with 1,000 bootstrap replicates (Felsenstein, 1985). Sequences with 2% or lower of genetic distance and clustered as a monophyletic group in the NJ tree will be assigned to an operational taxonomic unit (OTU) and considered as a species. One or two sequences from each OTU were selected to compare with nucleotide sequences from GenBank (National Center for Biotechnology Information, NCBI) with the standard nucleotide BLAST (Basic Local Alignment Search Tool). Eggs were associated with a given species only if the similarity values were 98% or greater.

Species diversity of spawning ground was characterized using Shannon–Wiener diversity index (Schoener, 1968). The Shannon–Wiener index is calculated as: H′=−ΣPi ln Pi

where pi is the proportion of species in total samples (egg numbers).

Similarities of species component between spawning grounds were evaluated by Jaccard’s similarity index (JI) (Real & Vargas, 1996) using the formula JI = a/(a +b +c), where ‘a’ is the number of species shared by two spawning ground, and ‘b’ and ‘c’ are the numbers of species specific in each spawning ground. Correlation between JI and the river-way distance was determined by the Mantel test with 1,000 random permutations performed in Arlequin v3.1 (Excoffier & Lischer, 2010).

Results

Species identification and diversity

Overall, 8,983 drifting eggs were recovered from the nine stations, of which 7,933 were sequenced successfully. All sequences were trimmed to the same length with 605 bp. A NJ tree, constructed to cluster species, revealed thirty-nine OTUs (Fig. 2). Comparing with sequences of adults and GenBank data, a total of 39 species were identified, representing 28 genera, four families, and two orders (Fig. 2, Table 1). The intraspecific and interspecific genetic distance range from 0 to 0.0288, and from 0.0308 (Botia superciliaris Günther, 1892—Botia reevesae (Chang, 1944) to 0.3038 (Glyptothorax sinensis Regan, 1908—Pseudolaubuca engraulis (Nichols, 1925), respectively (Showed in Supplemental Information 1). Approximately 64% of the species and 67% of eggs belonged to the Family Cyprinidae. In addition, 26% of the species and 28% of eggs were from the Family Cobitidae. The most commonly distributed species were the silver carp (H. molitrix), grass carp (C. idellus), silver gudgeon (S. argentatus) and Yichang gudgeon (G. filifer). No fish species were found in all sampled stations.

Figure 2 Neighbour-joining tree of 7,933 fish eggs using mtDNA cyt b gene.

The spawning ground with the highest species diversity was YJJ with 27 species identified, from 2014—2016. The next were YYD and YJZ, with 20 species found in both. The Shannon–Wiener (H) diversity index of these three spawning grounds were greater than 2.0, and the other five were under 1.9. Stations with the lowest species diversity were YPZ, having eight species.

Table 1 Species composition and diversity in nine sampled stations of the Yangtze River.

Orders	Families	Species	Stations & years	Total number of eggs	Ratio (%)	GenBank	
			YPZ	MYB	CCS	YJJ	YYD	YJZ	YSZ	YYW	XYT			No.	Identity, %	
			2016	2016	2016	2014	2015	2016	2015	2015	2015	2016	2015	2016	2016					
Cypriniformes	Cyprinidae	Hypophthalmichthys molitrix		2		28	39	97	169	28	1	2	8	6	11	391	4.929	EU315941	98–100	
Aristichthys nobilis				1			5		1			1	1	9	0.113	EU343733	99–100	
Ctenopharyngodon idellus		9		27	44	88	68	49	2	3	3	1	1	295	3.719	EU391390	99–100	
Mylopharyngodon piceus				8	7	12	13	7				1		48	0.605	EU979307	99-00	
Squaliobarbus curriculus							1	8			1			10	0.126	JX910141	98–100	
Elopichthys bambusa							2	23						25	0.315	KM196112	99–100	
Parabramis pekinensis							40	37	9		19	7	14	126	1.588	NC_022678	99–100	
Hemiculter bleekeri				7	9	23	50	113	274	19	43	52	213	803	10.122	KT361083	98–100	
Hemiculter leucisculus	40				1	9								50	0.630	AY089718	98–100	
Pseudolaubuca engraulis		15	1	84	320	407	1	4				3		835	10.526	NC_020462	99–100	
Pseudolaubuca sinensis			36		1	1	148	136	2					324	4.084	KY471356	99–100	
Culter mongolicus				1		3								4	0.050	AP009060	99	
Culter alburnus				1		6		1						8	0.101	KX829023	99–100	
Cultrichthys erythropterus						3								3	0.038	AF051859	99	
Coreius heterokon				49	30	93	5							177	2.231	NC_020041	99–100	
Rhinogobio typus		9		43	124	322	1	26	2		6	5		538	6.782	KU323963	98–100	
Saurogobio dabryi						1								1	0.013	NC_022845	99	
Hemiculterella sauvagei		5		2	5	15								27	0.340	KP316066	98–100	
Saurogobio gymnocheilus				1	28	35		4	21				9	98	1.235	KR362925	98–100	
Squalidus argentatus		1	31	1	38	63	13	111	36	5	3	20	171	493	6.215	KM654503	98–100	
Gobiobotia filifer		42	92	48	197	418	1	27	15	1	5	5	6	857	10.803	KP325413	98–100	
Xenophysogobio boulengeri	3	54	2			16			7					82	1.034	KM052390	98–100	
Xenoplysogobio nudicorpa		2												2	0.025	KM373519	98–100	
Pseudobrama simoni									20					20	0.252	KF537571	99–100	
Xenocypris argentea							41	10			4	1		56	0.706	AP009059	98–100	
Cobitidae	Leptobotia rubrilabris	1	1		1	5	15								23	0.290	AY625717	98–100	
Leptobotia tientaiensis							3	32	3	9	6		1	54	0.681	AY625725	98–100	
Leptobotia microphthalrna		1,303	1	1	11	365	1							1,682	21.203	NC_024049	99–100	
Leptobotia elongata	2	2	5	11	6	32								58	0.731	JQ230103	99–100	
Leptobotia taeniaps		3	9	3	5	19	3	19	4		2	3	1	71	0.895	KM386686	99–100	
Parabotia fasciata			4		18	13	10	30	2					77	0.971	AY625709	98–100	
Parabotia banarescui							2	4						6	0.076	AY625711	98–100	
Botia reevesae		1	2											3	0.038	KU954768	98–100	
Botia superciliaris	11	37	87	3	22	110								270	3.404	AY625704	98–100	
Paracobitis potanini		1												1	0.013	KP749475	99	
Balitoridae	Lepturichthys fimbriata	32	8	33	33	50	121		35	4			3		319	4.021	KJ830772	99–100	
Jinshaia sinensis	26	7		8	14	27								82	1.034	KJ739867	99–100	
Jinshaia.abbreviata		2												2	0.025	DQ105211	99	
Siluriformes	Sisoridae	Glyptothorax sinensis	2	1												3	0.038	KJ739617	99	
Total	117	1,505	303	361	974	2,314	577	704	403	39	100	108	428	7,933	100			
Species number	8	20	12	21	21	26	20	20	16	6	11	13	11	40				
Shonnon H	1.552	0.686	1.781	2.317	2.185	2.441	2.018	2.506	1.322	1.396	1.831	1.759	1.117	2.710				

Species similarity among sites

Pairwise Jaccard’s similarity index and river-way distances among sampled stations are listed in Table 2. There are only one or two identical species among 19 to 28 species between YPZ and the four stations in the Middle Yangtze River (YYD, YJZ, YSZ, and YYW) and XYT station in the Xiangjiang River, which obtained the least JI values, ranging from 0 to 0.0952. However, JI values among the four stations in the Middle Yangtze River had the highest values, ranging from 0.5652 to 0.7391.

Table 2 Fish Jaccard’s similarity index (JI) and river-way distances among sampled stations in the Yangtze River (Mantel test, R =  − 0.9003).

Stations	YPZ	MYB	CCS	YJJ	YYD	YJZ	YSZ	YYW	XYT	
YPZ	0	780	1,045	1,085	1,855	1,955	2,140	2,305	2,255	
MYB	0.3333	0	265	305	1,075	1,175	1,360	1,525	1,475	
CCS	0.2667	0.4545	0	150	920	1,020	1,205	1,370	1,320	
YJJ	0.2800	0.4687	0.3437	0	770	870	1,055	1,220	1,170	
YYD	0	0.2187	0.2800	0.4242	0	100	285	450	400	
YJZ	0.0370	0.2581	0.2800	0.4242	0.7391	0	185	350	300	
YSZ	0.0952	0.2593	0.3333	0.4333	0.5217	0.5652	0	165	115	
YYW	0.0909	0.2593	0.2500	0.3548	0.6500	0.6364	0.5500	0	210	
XYT	0	0.1852	0.1579	0.2667	0.4762	0.4286	0.6667	0.5625	0	

A strongly negative correlation was found between the JI and river-way distance (Mantel test, R =  − 0.9003).

Discussion

In previous studies, Cao et al. (2007) and Li et al. (2013) have shown that approximately 25 species of fish lay drifting eggs in the Yangtze River. Here we identified 39 species from captured drifting fish eggs in nine sampled stations using the mitochondrial Cyt b gene sequences. All obtained sequences had 98% identity with sequences in GenBank, indicating that reference data of the fish are available. Thus, fish eggs can successfully be identified to the species level. Intraspecific similarity of Cyt b gene sequence was greater than 98%, and interspecific distance higher than 2% for all these fish species. These studies indicate that mitochondrial sequence data are a power tool of species identification of drifting eggs in the Yangtze River, and may also contribute to identify rare or endangered species of fish. For example, some fish such as Saurogobio gymnocheilus Lo, Yao & Chen, 1998, B. reevesae, Paracobitis potanini Günther, 1896, and Jinshaia abbreviata Günther, 1892, have had a low catch rate in recent years, but were detected in one or two spawning grounds in this study (Table 1). This observation may suggest that there are also smaller populations living in the river. However, two common fish species that lay drifting eggs in the Upper Yangtze River, C. guichenoti and Rhinogobio ventralis Sauvage & Dabry de Thiersant, 1874 (Xiong et al., 2014; Liu et al., 2010; Xiong et al., 2016), were not found in this study. This may indicate that their spawning grounds were not in our monitoring stations. In addition, two recorded Leuciscin carps laying drifting eggs, Ochetobius elongatus Kner, 1867 (still found in Pearl River) and; Luciobrama macrocephalus Lacepède, 1803 in the middle Yangtze River, were also not found from analysis of eggs. Since they have been not captured in recent years it is possible that these species are endangered to extinction in the Yangtze River.

Several species from the Culters genus were found in the collected eggs, such as Culter alburnus Basilewsky, 1855, Culter mongolicus Dybowski, 1872, and Cultrichthy s erythropterus Basilewsky, 1855. Interestingly, these are not typical fish that lay drifting eggs. They generally spawn weakly viscous eggs among water plants or in river gravel and can also spawn in still water (Cao et al., 2007). They were detected in the YJJ station due to falling off from plants or gravels by river torrents. G. sinensis which is distributed in the Yangtze River and its tributaries and lakes, also produces viscous eggs not drifting, and their eggs usually adhere to stones in the water (Ding, 1994; Chu, Zhen & Dai, 1999). Two eggs of G. sinensis were found in the YPZ station, where the current velocity could be greater than 2 m/s (Xu, Wang & Fu, 2013).

The negative correlation between species similarity and river way distance reflected the diversity of ecological conditions in the Yangtze River. The nearer river stations have similar ecological characteristics, such as temperature, current velocity, and water flow, which attract more of the same fish to spawn. There are distinct differences in hydrology between the middle and upper reaches of the Yangtze River. The river is narrow, swift, and with a steep fall in elevation in the upper reaches compared with the wider, slower, and gradual fall in the middle reaches. Ecological differences shape the specificity of fish in the spawn ground. Six species, including Squaliobarbus curriculus Richardson, 1846, Elopichthys bambusa Richardson, 1845, Parabramis pekinensis Basilewsky, 1855, Pseudobrama simoni Bleeker, 1864, Xenocypris argentea Bleeker, 1871 and Parabotia banarescui Nalbant, 1965 are only found in the YYD, YJZ and/or YSZ, and YYW stations, consistent with their endemic presence in the middle Yangtze River. The upper Yangtze River involved fourteen species, including S. gymnocheilus, Hemiculterella sauvagei Warpachowski, 1888, Xenophysogobio boulengeri Tchang, 1929, Xenophysogobio nudicorpa Huang & Zhang, 1986, Leptobotia rubrilabris Leptobotia rubrilabris,Dabry de Thiersant, 1872, Leptobotia microphthalma Fu & Ye, 1983, Leptobotia elongata Bleeker, 1870, B. reevesae, B. superciliaris, P. potanini, Jinshaia sinensis Sauvage & Dabry de Thiersant, 1874, and J. abbreviate. Adults of L. elongate, the largest fish in the Family Cobitidae, were also found in the middle Yangtze River, but no eggs were detected in stations there. Larvae of L. elongate born upstream should have crossed the Three Gorges Dam to the middle reaches of the river.

Species diversity of the YJJ station was shown to be the richest, and it harbored not only some importantly economic fish, such as FDC, also some unique fish, such as S. gymnocheilus (Table 1). This river reach is a very important spawning ground for fish laying drifting eggs. The numbers of species in the tributary river is less comparing with the Main River, but the rivers could be important spawning grounds for particular species. For example, the number of the small eye loach, L. microphthalma accounted for more than 86% of the total eggs in the MYB station, where in down reaches of Minjiang River. In the CCS station, located in the Chishui River, the species and egg numbers of species in the Family Cobitidae accounted for 50% and 36% of the total captures, respectively.

Though spawning and egg development of most of the fish identified in this study rely on running water, dependency levels were different. For example, FCD are apt to spawn in sudden floods (Duan et al., 2009; Li, Cai & Tang, 2011). However, spawning activity of G. filifer is negatively correlated with water discharge and water velocity (Tian et al., 2017). Understanding the composition of fish species at each spawning ground helps to assess the impact of environmental changes on fish, and molecular methods could be a useful tool for conducting this task effectively.

Our study presented a highly effective approach for rapid species identification of drifting eggs in Yangtze River, and it should due to reference data obtained in Genbank. A small percentage of collected eggs were not sequenced (approximately 11%), which can be attributed to several errors along the path from collection to sequencing, such as improper egg preservation, DNA extraction and PCR failure. For large scale survey, the lab time as well as amplification and sequencing costs cannot to be ignored. Using high-throughput pipelines as an automatization of egg sorting, and a 96-well system for DNA extraction and for PCR amplification (Hofmann et al., 2017) and/or multiplex arrays (Gleason & Burton, 2012) is recommended. Another problem using a single gene of mtDNA for species identification is the presence of heteroplasmy or incomplete lineage sorting within or among some species, which lead to aborted identification at species-level (Hubert et al., 2008; April et al., 2011; Shen et al., 2016). Using more loci, for example the COI, control region and ITS of rDNA, can improve the success rate of identification.

Conclusions

Mitochondrial DNA sequence is a powerful and effective tool for identifying species of drifting fish eggs in Yangtze River. Species diversity in the main river is higher than that in tributaries; however, some fish species prefer to spawn in tributaries.

Supplemental Information

Supplemental Information 1 The intraspecific and interspecific genetic distance based on Kimura 2-parameter model in this study

Click here for additional data file.

Supplemental Information 2 Sequence data, identification results and neighbor-joining phylogeny

Click here for additional data file.

We thank International Science Editing (http://www.internationalscienceediting.com) for editing this manuscript.

Additional Information and Declarations

Competing Interests

Author Contributions

Data Availability

The authors declare there are no competing interests.

Mingdian Liu conceived and designed the experiments, performed the experiments, analyzed the data, prepared figures and/or tables, authored or reviewed drafts of the paper, approved the final draft.

Dengqiang Wang performed the experiments, analyzed the data, contributed reagents/materials/analysis tools, prepared figures and/or tables, authored or reviewed drafts of the paper.

Lei Gao performed the experiments, prepared figures and/or tables.

Huiwu Tian performed the experiments, contributed reagents/materials/analysis tools.

Shaoping Liu and Daqing Chen conceived and designed the experiments.

Xinbin Duan conceived and designed the experiments, authored or reviewed drafts of the paper, approved the final draft.

The following information was supplied regarding data availability: The raw data are provided in the Supplemental File.

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
