# Peer review of "Species diversity of drifting fish eggs in the Yangtze River using molecular identification"

_PeerJ, doi:10.7717/peerj.5807_

## Round 0.1 · original submission · Minor Revisions

Though, species identification through barcoding fish eggs is not new, the study is informative! the two reviewers' raise several important points to enrich the manuscript. please address all the suggestions and queries clearly.

·

Basic reporting

In this study Liu et al. analyzed an incredible high number of freshwater fish eggs of the Yangtze River of China using a mitochondrial gene fragment (Cyt bB). I think that the analysis and results are interesting and likely helpful for other researchers in this research field, but I favor a somewhat more broader and comprehensive discussion of the application of molecular data for the identification of fish eggs. Take a look at Hofmann et al (2014) (Journal of Applied Ichthyology 33: 925-932) and other similar studies. The authors should also include a discussion of modern molecular approaches in order to assess species biodiversity of a habitat, e.g., the use of high-throughput sequencing technologies (meta-barcoding), proteomics (MALDI-TOF), and others. In this context I miss a comprehensive discussion of possible pitfalls of using mtDNA in terms of species delineation and specimen identification, e.g., the existence of incomplete lineage sorting, heteroplasmy etc. as part of the introduction or discussion. Is it possible to differentiate the analyzed fish eggs by the means of morphological traits? Currently, no information about this important aspect is given.
Acceptable English is used throughout, except for some minor grammatical mistakes. However, the current phrasing of the abstract should be rephrased and improved. In addition, please check the manuscript carefully for typos, e.g. line 16 (identified instead of identify) or 42 (Hypophthalmmichthys nobilis instead of Hypophthalmmichthys, nobilis). There are various other typos existing, including missing blanks or double blanks.

Detailed comments

Line 40: Please provide name of first descriptors of the studied species here and in the following

Line 108: Change “Cyt b” to “Cyt b (b in italics)” here and in the following

Line 108-115: I feel that the space lining of this part is not correct

Line 109: Did you have published the used primer before? In such a case you should only provide the primer name and reference instead of listing up all nucleotides. If both primers are new, however, you should re-arrange them in a triplet code: GAC TTG AAG …

Line 111: Just a comment: Think about using a smaller PCR mix (25 µl instead of 50 µl) in order to save resources.

Line 116: Have you checked your sequences for the presence of pseudo genes?

Line 118: Which company was employed for sequencing your amplicons?

Line 119: Currently, no accession numbers are provided. However, this is essential for publication.

Line 122: Even if you have analyzed protein-coding gene sequences of more or less closely related species and not taxa from different phyla, Clustal W and Clustal X should not be used to align sequence data any more. Other programs, e.g. Muscle, MAFFT or Clustal Omega are more useful and adequate

Line 123: Provide a reference for the K2P model

Line 124: Think about updating your MEGA version; the actual version is 7. It has also implemented Muscle (see above)

Line 124-125: Please give references for the neighbor joining approach (> Saitou & Nei 1987) and the non-parametric bootstrap analysis (> Felsenstein 1985)

Line 139: Replace “Arliquin” by “Arlequin”

Line 144: The total number of analyzed eggs was 8983, but only 7933 eggs were studied. Why?

Line 144 following: Whereas you mentioned reconstructing a NJ topology in the M&M paragraph, no figure is shown. Why?

Line 144 following: In think you should highlight your results a little bit more than refer on table 1. This is also true for the discussion. For example, you can provide percentage values for the most important values

Line 173: Replace “molecular methods” by “mitochondrial sequence data”

Line 186: Culters in italics

Line 207-209: Rephrase this sentence

Line 270-274: Whereas you did not use DNA barcoding, you cite two studies of Hebert and co-authors. However, they do not appear within the manuscript. This is also true for Hajibabaei et al. and various others. Check your reference paragraph carefully.

Experimental design

The research question is relevant and well explained. In terms of using DNA sequence data to identify specimens, however, the analysis of a partial fragment of the CO1 gene has become the most popular approach by far (DNA barcoding sensu Hebert). This is also true for fish species. So, can you explain why you have used cytochrome b instead of CO1? Furthermore, I miss the NJ topology of the analyzed taxa.

Validity of the findings

I think that the authors should highlight their results a little bit more than to refer on table 1. This is also true for the discussion. For example, they can provide percentage values for the most important values. What about inter- and intraspecific distances? Such information will help to improve the manuscript.

·

Basic reporting

The manuscript is well written in English and clear throughout the text. The background/context provided in the manuscript for the study conducted is good, and the figures and the tables, containing relevant data and that allow the authors to reach their main conclusions, were provided. The raw data is also shared. Literature references provided are enough. The article followed a clear structure with standard sections and sub-sections. Although the subject is not completely new (fish species identification through barcoding of DNA extracted from eggs), it is highly relevant and provides a clear example where species identification greatly benefits from the use of DNA-based tools.

Experimental design

The authors conducted a very interesting study about the molecular identification, based on DNA barcoding, of fish species using the eggs spawned and collected from the Yangtze River and its tributaries (China), during a period of 2 years. The research question is well defined in the manuscript and the background and context provided reveals its high relevance. The methods were adequate to address the questions of the current study, as well as the data analyses conducted, and enough details were provided. The permissions needed to sample the eggs were also obtained before the study. There is just two points for which the authors should give more details: 1) why did the authors used Mitochondrial (mt) cyt b gene marker instead of Mitochondrial COI marker, since COI is the reference DNA barcode used for animal species identification (thus reference databases would be more complete with COI barcodes belonging to fish species) and 2) when the authors state in the data analysis sub-section that all sequences were trimmed to the same length; the authors should provide the final length of the sequences. One negative point of the study is that there were stations that were sampled only in 2016.

Validity of the findings

The authors are very straightforward in presenting their results, which is very good. The discussion is well conducted, but please verify again the nomenclature of the fish species since I have found some errors in the names of some species. The conclusions were appropriately stated and the importance of using DNA-based tools was proved by using this particular case. However, one point that is not very clear is if the samplings were just conducted once, during the period of May to July, or more than once. The results would be more reliable if the eggs counts provided were based in different sampling points between May and July (replicates), for each station.

Additional comments

Minor comments:
In Table 1 correct "total" by "Total number of eggs"; "speices" by "species" and "Shonnon J" by "Shannon H"
I would suggest that you change Table 2 legend to "Fish Jaccard’s similarity index (JI) and river-way distances among sampled stations in the Yangtze River (Mantel test, R=-0.9003)"
Other minor edits were directly made in the document that I send attached (edits made are in changetrack).

---

## Round 0.2 · Minor Revisions

Just few things need to revised as per reviewers' queries! Hopefully the manuscript then will be in a acceptable form! good work!

·

Basic reporting

It is nice to see that the authors accepted all comments and/or remarks. However, some minor concerns still remain. In this context, I still miss a broader and comprehensive discussion of the application of molecular data for the identification of fish eggs. Which methods have been used? Furthermore, the authors should carefully differentiate between the use of DNA barcodes (sensu Hebert et al. 2003) and the application of other markers for identification. Currently, all is mixed without structure. Furthermore, there are still various typos existing (e.g., missing blanks). I tried to get most, but please check the whole manuscript carefully. I also still miss a summary and discussion of the observed inter- and intraspecific molecular distances of the analyzed fish species. This is quite important.

Abstract: Whereas some work has been done here, I still think that current phrasing of the abstract should be rephrased and improved.

Line 36: Please provide name of first descriptors of the studied species here and in the following (e.g., Ctenopharyngodon idella Valenciennes, 1844). They are still missing but should be added.

Line 117-120: I still think that the space lining of this part is not correct.

Please see some specific comments made via sticky notes on the PDF file of the manuscript.

Experimental design

As already mentioned, the research question is relevant and well explained. As part of thier reply the authors clearly explained why they used cyt b instead of CO1. The (detailed) NJ tree is part of the electronic supplement. However, it would be nice to add a somewhat simplified NJ tree with collapsed species (see MEGA tool: compress/expand subtree after a phylogenetic analysis) to the manuscript.

Validity of the findings

No more comments (see above).

·

Basic reporting

This is a revised version of a manuscript that I have previously revised. The authors have carefully addressed the two reviewers’ suggestions and concerns, yielding an improved version of the manuscript.

Experimental design

In the first version of the MS, there were some experimental issues that arose, but that the authors have clarified in this revised version.

Validity of the findings

The authors have clarified less clear issues found in the previous version of this MS.

Additional comments

Other minor edits were directly made in the document that I send attached (edits made are in changetrack).

---

## Round 0.3 · accepted · Accept

Congratulations!
Thanks for addressing the issues raised by the reviewers'.
Hope your contribution will enrich the field.
Stay with PeerJ!

#